# The association between temperature and alcohol- and substance-related disorder hospital visits in New York State

Robbie M. Parks [1,2✉], Sebastian T. Rowland[1], Vivian Do[1], Amelia K. Boehme[3,4], Francesca Dominici[5], Carl L. Hart[6,7] & Marianthi-Anna Kioumourtzoglou[1]

**Abstract**

**Background** Limited evidence exists on how temperature increases are associated with hospital visits from alcohol- and substance-related disorders, despite plausible behavioral and physiological pathways.

**Methods** In the present study, we implemented a case-crossover design, which controls for seasonal patterns, long-term trends, and non- or slowly-varying confounders, with distributed lag non-linear temperature terms (0–6 days) to estimate associations between daily ZIP Code-level temperature and alcohol- and substance-related disorder hospital visit rates in New York State during 1995–2014. We also examined four substance-related disorder sub-causes (cannabis, cocaine, opioid, sedatives).

**Results** Here we show that, for alcohol-related disorders, a daily increase in temperature from the daily minimum ($-30.1\,°C$ ($-22.2\,°F$)) to the 75th percentile ($18.8\,°C$ ($65.8\,°F$)) across 0–6 lag days is associated with a cumulative 24.6% (95%CI,14.6%–34.6%) increase in hospital visit rates, largely driven by increases on the day of and day before hospital visit, with an association larger outside New York City. For substance-related disorders, we find evidence of a positive association at temperatures from the daily minimum ($-30.1\,°C$ ($-22.2\,°F$)) to the 50th percentile ($10.4\,°C$ ($50.7\,°F$)) (37.7% (95%CI,27.2%–48.2%)), but not at higher temperatures. Findings are consistent across age group, sex, and social vulnerability.

**Conclusions** Our work highlights how hospital visits from alcohol- and substance-related disorders are currently impacted by elevated temperatures and could be further affected by rising temperatures resulting from climate change. Enhanced social infrastructure and health system interventions could mitigate these impacts.

**Plain language summary**

We investigated the relationship between temperature and hospital visits related to alcohol and other drugs including cannabis, cocaine, opioids, and sedatives in New York State. We found that higher temperatures resulted in more hospital visits for alcohol. For other drugs, higher temperatures also resulted in more hospital visits but only up to a certain temperature level. Our findings suggest that rising temperatures, including those caused by climate change, may influence hospital visits for alcohol and other drugs, emphasizing the need for appropriate and proportionate social and health interventions, as well as highlighting potential hidden burdens of climate change.

[1] Department of Environmental Health Sciences, Mailman School of Public Health, Columbia University, New York, NY, USA. [2] The Earth Institute, Columbia University, New York, NY, USA. [3] Department of Neurology, Columbia University Medical School, New York, NY, USA. [4] Department of Epidemiology, Mailman School of Public Health, Columbia University, New York, NY, USA. [5] Department of Biostatistics, T.H. Chan School of Public Health, Harvard University, Boston, MA, USA. [6] Department of Psychology, Columbia University, New York, NY, USA. [7] Department of Psychiatry, Columbia University, New York, NY, USA. ✉email: robbie.parks@columbia.edu

Based on a Substance Abuse and Mental Health Services Administration (SAMHSA) survey in 2019 in the United States, 139.7 million people had recently drunk alcohol, with at least 25.8 million consuming illicit drugs[1]. Most substance users do so in moderation[1]. However, a substantial minority of individuals suffer from related disorders, defined as when recurrent use of alcohol or other substances causes clinically significant impairment[2], and require intervention, including inpatient treatment. Many individuals requiring intervention are regular substance users[1]. One in eight deaths in working-age Americans are estimated to be associated with excessive alcohol use[3]. In recent decades, there has been an increasing trend of heavy episodic drinking and alcohol-related morbidity and mortality in the United States, particularly in middle-aged to older adults[4].

Warm and cold weather events are both significant issues related to public health, serving as crucial catalysts for the implementation of adaptation strategies to combat climate change. Evaluations of the impacts of weather and climate on health, including the broader implications of global climate change, have predominantly concentrated on infectious and parasitic diseases, as well as chronic conditions affecting the cardiovascular and respiratory systems[5,6]. Recent work has focused on mental health-related mortality outcomes in the United States, such as suicide and violence[7,8]. There are plausible direct behavioral and physiological pathways for a relationship between changes in temperature and alcohol- and substance-related disorders; increased consumption in warmer weather, more perspiration, and temperature-dependent efficacies of certain substances, such as opioids, may all contribute to changes in alcohol and substance use and how a human body reacts to their consumption[9]. There are also indirect pathways between rising temperatures from climate change and worsening mental health outcomes that could lead to the increasing use of alcohol and other substances, such as deteriorating social fabric and widening inequality[10,11].

Limited previous studies of alcohol- and substance-related disorders in relation to temperature exist. A near-linear association was found between alcohol or drug poisonings discovered during unhoused rescue missions and temperature in Hamburg, Germany[12], while another study of alcohol use disorders in Paris, France found a correlation of 0.55 between weekly alcohol use disorders and mean temperature[13]. However, neither study adequately controlled for confounding bias, including season. Heatwaves in Hanoi, Vietnam were associated with increases in admissions from mental disorders in general, but not from psychoactive substance use[14]. Substance abuse-related mental illness emergency hospital visits in Toronto, Canada were positively associated with higher temperatures[15]. A study leveraging information on Optum hospital visits, a selected sub-set of total United States hospital visits based on insurance records, identified a positive association with temperature and substance-related disorders[16]. Studies of rats have found that high ambient temperatures impact the acquisition of 3,4-methylenedioxymethamphetamine (MDMA) and methamphetamine, as well as dopamine self-administration[17–20]. Nevertheless, there remains an overall knowledge gap in consistently and comprehensively quantifying how temperature is associated with alcohol- and substance-related hospital visits.

The aim of this study was to evaluate (a) how daily temperature was associated with hospital visits due to alcohol- or substance-related disorders (including alcohol, cannabis, cocaine, opioids and sedatives), and (b) how this association varied by location, age group, sex, and social vulnerability, using daily ZIP Code-level hospital visit data obtained from hospitals in New York State, the fourth largest state by population in the United States[21].

We found that (i) an increase in temperature 0–6 days before hospital visit was associated with higher hospital visit rates for both alcohol- and substance-related disorders, and that (ii) the association was greatest for substance-related disorders outside New York City.

## Methods

This study was approved by the Institutional Review Board at the Columbia Mailman School of Public Health and was classified as exempt from needing to obtain Informed Consent (Protocol IRB-AAAR0877).

**Study Population.** Hospital records were obtained across NYS from 1995 to 2014 from the New York Department of Health Statewide Planning and Research Cooperative System (SPARCS) (https://www.health.ny.gov/statistics/sparcs/). SPARCS is an administrative dataset collected from all non-military acute care facilities in NYS, covering ~98% of all NYS hospital visits; as of 2015, SPARCS included 222 acute care facilities[22]. For each admission record, International Classification of Diseases, Ninth Revision, Clinical Modification (ICD-9-CM) diagnosis codes were obtained, along with patient residential ZIP Code, date of admission, age, and sex.

**Outcomes.** Alcohol- and substance-related disorder cases were identified from the first four ICD-9-CM diagnostic position codes in each admission record. Both inpatient and outpatient admissions were included. Classifications were based on the Clinical Classifications Software (CCS) algorithm[23], commonly used in epidemiologic studies to group ICD codes into clinically-meaningful categories (Supplementary Table 1)[24–26]. Substance-related disorder records were further subdivided. This resulted in two broad causes (alcohol-related disorders, substance-related disorders) and four specific substance-related sub-causes (cannabis, cocaine, opioids, sedatives). For each cause, an admission was counted as a case if it included at least one matching code in the four ICD-9-CM codes, such that a single admission could be attributed to several causes.

**Exposure.** Daily average temperature, specific humidity, and pressure were obtained from the North American Land Data Assimilation System, NLDAS-2 Forcing[27], with full space and time coverage over the study period. NLDAS-2 estimates hourly mean weather values within 0.125° grids (~11 km × 14 km in NYS). Similar to previous work[22,24,28,29], weather variable grid daily averages were intersected with census tract-level population from 2010 US Census data. Population-weighted averages were then computed at the ZIP Code Tabulation Area (ZCTA) level, a consistent geographic representation of ZIP Codes (https://www.census.gov/programs-surveys/geography/guidance/geo-areas/zctas.html), referred to as ZIP Code hereafter (Supplementary Fig. 13). Relative humidity (RH) was calculated from temperature, specific humidity, and pressure (Supplementary Fig. 14)[30].

**Covariates.** Data on social vulnerability in NYS by census tract were used from the Centers for Disease Control and Prevention (CDC) Social Vulnerability Index (SVI) for 2014 (https://www.atsdr.cdc.gov/placeandhealth/svi/data_documentation_download.html). The SVI incorporates data from the US Census on socioeconomic status; household composition and disability; minority status and language; and housing type and transportation to determine the relative social vulnerability of every census tract in NYS[31]. A census tract's SVI value indicates the relative vulnerability of every NYS census tract compared with every other NYS census tract, ranking from 0 (lowest vulnerability in

the state) to 1 (highest vulnerability in the state). To obtain ZIP Code-level SVI values, the 4,903 census tract SVI values were area-weighted into 1,794 ZIP Codes. The ZIP Codes were divided into SVI tertiles (low vulnerability to high vulnerability, 1 to 3; Supplementary Fig. 15). Each SVI tertile contained 598 ZIP Codes. The same SVI tertile values were used for each ZIP Code throughout analyses.

**Statistical analysis**. A time-stratified case-crossover design was used, commonly used for analyzing associations with short-term exposures[32,33]. In this design, temperature of the day of hospital visit and relevant preceding days (case period) are compared with the temperature of sets of days where the hospital visit did not occur (control periods). This study design utilizes every single hospital visit, not only those during periods of high temperatures. Comparing hospitalized individuals to themselves during other periods when they were not hospitalized eliminates confounding due to factors that vary across individuals. A conditional logistic regression[33] was used to quantify the association between daily average temperature and hospital visit rates, coupled with distributed lag non-linear model (dlnm) terms to estimate cumulative associations prior to the hospital visit[34]. Cumulative associations were chosen to represent the total association in a parsimonious way. Six days' cumulative association prior to hospital visit was chosen to include the most acute associations from high temperatures[35], while also maximizing power by not overlapping case and control periods. The cumulative association of only the temperature on the day of and day before was also estimated. Relative humidity was adjusted for, also including distributed lag terms, equivalent to the structure of the temperature terms. Specifically, via a logit function, the log-odds of hospital visit were modelled as follows:

$$logit\left[\Pr\left(Y_{ci}=1\right)\right] = \alpha_c + \sum_{l=0}^{6} s(T, df)_{lci} + \sum_{l=0}^{6} s(RH, df)_{lci}, \quad (1)$$

where $Y_{ci}$ denotes whether subject $i$ in matched stratum $c$ was hospitalized, i.e., $c$ represents a group of a case and its matched controls; $\alpha_c$ the matched stratum-specific intercepts (not estimated in conditional logistic models); $s(T, df)_{lci}$ the lag-specific natural spline terms as part of the dlnm terms for temperature; and $s(RH, df)_{lci}$ the lag-specific natural spline terms as part of the dlnm terms for relative humidity. To select the optimal fit for the non-linear dlnm terms, models for alcohol-related disorders and substance-related disorders were fit separately using a variety of plausible degrees of freedom (dfs) to model the lag-specific exposure – response function ($df_{var}$), as well as the function of the association over the examined lags ($df_{lag}$). A range of 2 to 5 for $df_{lag}$ were considered, along with between 3 and 4 for $df_{var}$. The optimal values were selected by choosing the combination of $df_{lag}$ and $df_{var}$ with the lowest Akaike Information Criteria (AIC) values[36]. The models with lowest AIC values for both causes were $df_{lag} = 4$ and $df_{var} = 3$.

In addition to the main analyses investigating all hospital visits together for each cause and sub-cause (alcohol-related disorders, substance-related disorders, cannabis, cocaine, opioids and sedatives), further assessment was made of whether estimated effects varied by location (NYC or not NYC), sex (female or male), age group (0–24 years, 25–44 years, 45–64 years or 65+ years), or by SVI tertile (low vulnerability to high vulnerability, 1 to 3), by conducting stratified analyses, using the same model as described above.

Unless stated otherwise, results are presented as cumulative percentage change in hospital visit rates were each of the lag days (0 to 6 days before) at the quoted temperature (e.g., the 75th

percentile; 18.8 °C (65.8 °F)) relative to −30.1 °C (−22.2 °F), the New York State daily minimal temperature throughout the study period, appropriate for case crossover model output[32]. Statistical analyses were conducted using the R Statistical Software, version 4.1.1[37], and dlnm, version 2.4.2[38].

**Sensitivity analyses**. The sensitivity of the results to potential confounding by relative humidity was assessed by removing the relative humidity terms from the models.

**Reporting summary**. Further information on research design is available in the Nature Portfolio Reporting Summary linked to this article.

## Results

**Hospital visits**. There were 717,798 total hospital visit records in New York State for alcohol-related disorders and 794,305 for substance-related disorders during the study period (1995–2014). Admissions with missing, incomplete, or inaccurate records of sex, age, dates of admission, or residential ZIP Codes were excluded (16.4% of alcohol-related disorder and 9.2% of substance-related disorder hospital visits). This left 671,625 complete hospital visit records for alcohol-related disorders and 721,469 for substance-related disorders (Table 1). Across sub-causes of substance-related disorders, total complete hospital visit records ranged from the highest for opioids (275,707) to lowest for sedatives (50,068). Across every cause, the age group with largest proportion of hospital visits was 25–44 years, from 46% of alcohol-related disorder hospital visits up to 61% of cocaine hospital visits. Males made up the majority of hospital visits across all causes, from 53% in sedatives to 63% in alcohol-related disorders. Most hospital visits were in-patient, from 68% of cannabis hospital visits to 87% of sedative hospital visits. Most hospital visits were also not in NYC, from 53% of hospital visits for cocaine and opioids, to 67% for cannabis.

In New York State, the number of alcohol- and substance-related disorders varied by ZIP Code (Fig. 1). The maximal total number of hospital visits in a single ZIP Code was 6,479 for alcohol-related disorder hospital visit in Troy (12180) and 8,026 substance-related disorder hospital visits in East Harlem (10029), though the observed patterns can be driven by differences in demographic structure of the population. Many cases were concentrated in urban environments. Overall, there was a high correlation (R = 0.98) between total numbers of hospital visits for both alcohol- and substance-related disorders across all ZIP Codes.

Over time, the number of alcohol- and substance-related disorder hospital visits increased across females and males, as well as in NYC and not NYC (Fig. 2). There were more substance-related disorders than alcohol-related disorders throughout the study period. Trends in increased substance-related disorder hospital visits over time were driven by increases in cannabis and opioids, with increases then slight decreases for cocaine and sedatives (Supplementary Fig. 1).

**Association of temperature with total hospital visits**. For alcohol-related disorder hospital visits, an increase in temperature from the period minimum (−30.1 °C (−22.2 °F)) to the 75th percentile (18.8 °C (65.8 °F)) across 0–6 lag days was associated with a cumulative 24.6% (95%CI, 14.6%–34.6%) increase in hospital visit rates, and an increase from the average to the 90th percentile (22.7 °C (72.9 °F)) was associated with a cumulative 25.6% (95%CI, 15.4%–35.7%) increase (Fig. 3). Overall, there was a near-linear positive association between temperature and alcohol-related disorders hospital visits across most of the

| Characteristic | Alcohol-related disorders, N = 671,625[a] | Substance-related disorders, N = 721,469[a] | Cannabis, N = 139,240[a] | Cocaine, N = 228,989[a] | Opioids, N = 275,707[a] | Sedatives, N = 50,068[a] |
|---|---|---|---|---|---|---|
| **Age group (years)** | | | | | | |
| 0–24 years | 58,320 (8.7%) | 120,077 (17%) | 52,307 (38%) | 20,097 (8.8%) | 34,964 (13%) | 5,740 (11%) |
| 25–44 years | 310,415 (46%) | 373,214 (52%) | 65,870 (47%) | 139,053 (61%) | 145,307 (53%) | 24,812 (50%) |
| 45–64 years | 271,144 (40%) | 209,459 (29%) | 20,534 (15%) | 68,280 (30%) | 89,615 (33%) | 17,203 (34%) |
| 65+ years | 31,746 (4.7%) | 18,719 (2.6%) | 529 (0.4%) | 1,559 (0.7%) | 5,821 (2.1%) | 2,313 (4.6%) |
| **Sex** | | | | | | |
| Female | 246,404 (37%) | 303,549 (42%) | 55,322 (40%) | 89,734 (39%) | 116,180 (42%) | 23,469 (47%) |
| Male | 425,221 (63%) | 417,920 (58%) | 83,918 (60%) | 139,255 (61%) | 159,527 (58%) | 26,599 (53%) |
| **Admission type** | | | | | | |
| In-patient | 510,449 (76%) | 535,334 (74%) | 94,756 (68%) | 191,250 (84%) | 229,442 (83%) | 43,352 (87%) |
| Out-patient | 161,176 (24%) | 186,135 (26%) | 44,484 (32%) | 37,739 (16%) | 46,265 (17%) | 6,716 (13%) |
| **Where in NYS** | | | | | | |
| NYC | 261,944 (39%) | 305,900 (42%) | 45,986 (33%) | 108,318 (47%) | 129,607 (47%) | 21,799 (44%) |
| Not NYC | 409,681 (61%) | 415,569 (58%) | 93,254 (67%) | 120,671 (53%) | 146,100 (53%) | 28,269 (56%) |

**Table 1 Demographic characteristics for hospital visits in New York State for 1995–2014.**

[a]n (%).

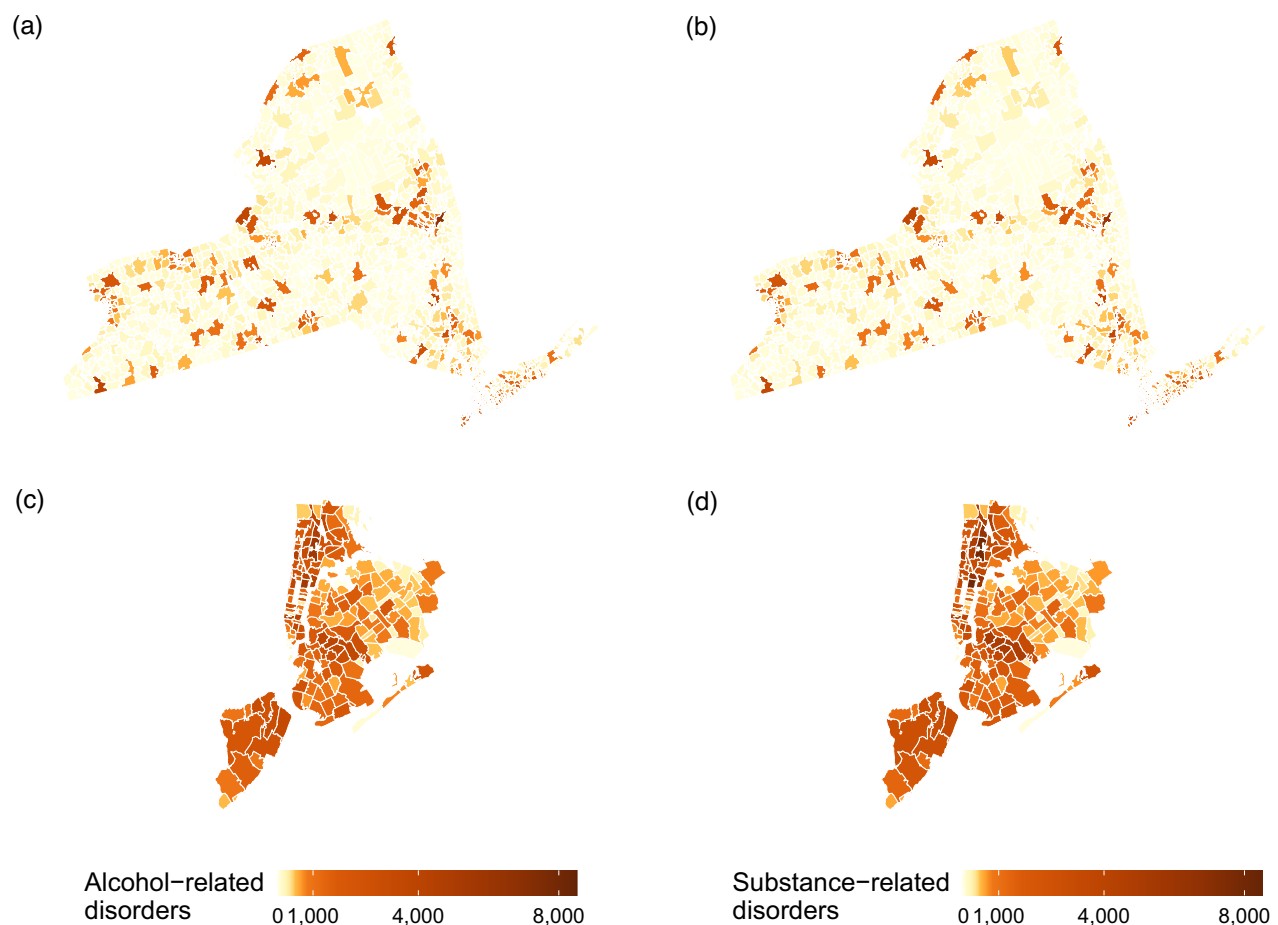

**Fig. 1 Map of total hospital visits by ZIP Code Tabulation Area for 1995–2014.** Cause- and location-specific maps of: (**a**) alcohol-related disorder hospital visits in New York State (n = 671,625); (**b**) substance-related disorder hospital visits in New York State (n = 721,469); (**c**) alcohol-related disorder hospital visits in New York City (n = 261,944); and (**d**) substance-related disorder hospital visits in New York City (n = 305,900).

temperature distribution. For substance-related disorders, we found an association between increases in temperatures from the period minimum (−30.1 °C (−22.2 °F)) to the 75th percentile (18.8 °C (65.8 °F)), with a (38.8% (95%CI, 28.7%–48.8%) increase. Overall, there was a near-linear positive association between

temperature and substance-related disorders hospital visits below the mean temperature and no evidence of an additional increase beyond, with increase from the period minimum to the 90th percentile (22.7 °C (72.9 °F)) of 39.0% (28.8%–49.2%). Cannabis-related admissions followed the overall substance-related disorder

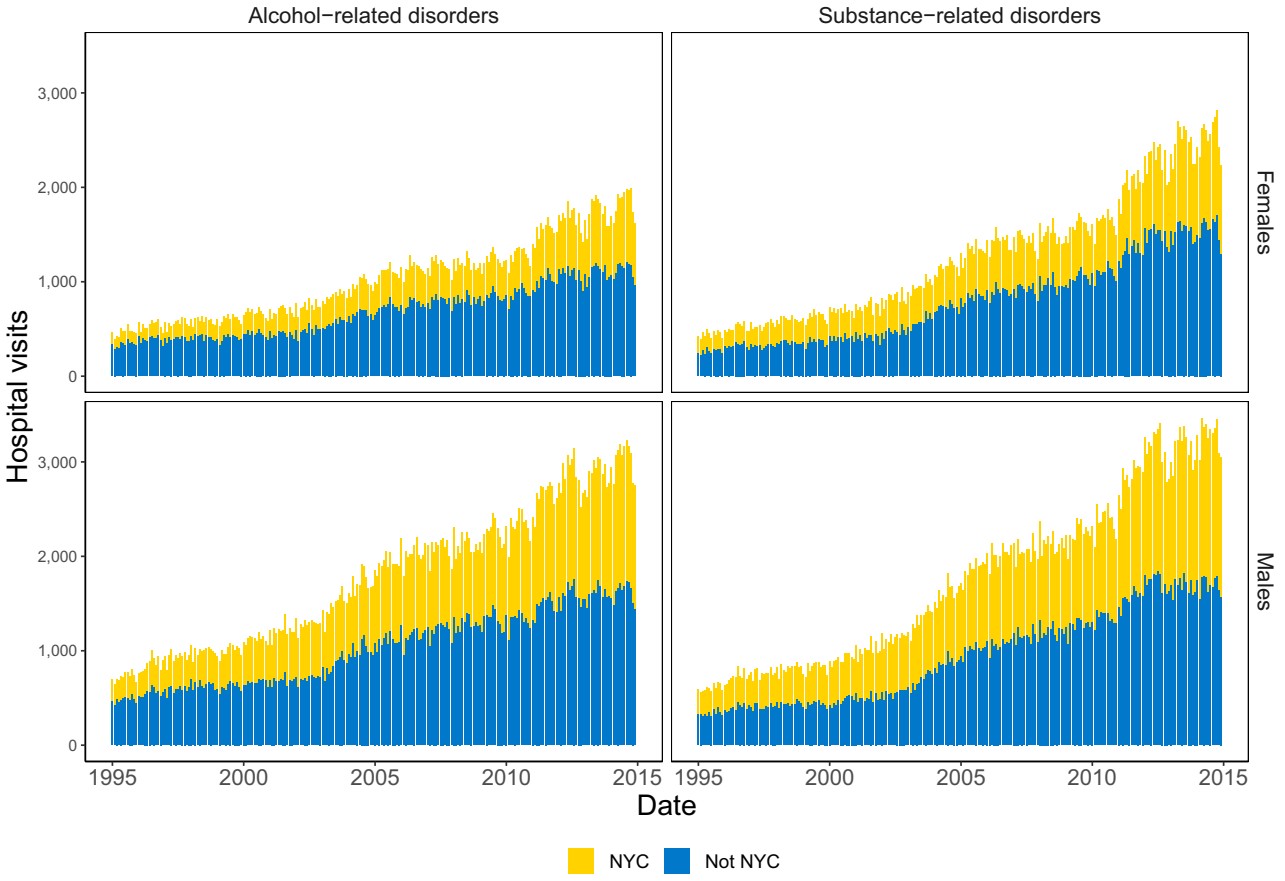

**Fig. 2 Monthly hospital visits, by cause, sex, and location in New York State for 1995–2014.** Causes include alcohol- ($n = 671,625$) and substance-related ($n = 721,469$) disorders. New York City is yellow, and the rest of the state is blue.

association patterns (e.g., an increase in temperature from daily minimum ($-30.1\,°C$ ($-22.2\,°F$)) to 75th percentile ($18.8\,°C$ ($65.8\,°F$)) across 0–6 lag days was associated with a cumulative 42.6% (95%CI, 21.6%–63.7%) increase in hospital visit rates) (Supplementary Fig. 2). Cocaine-related admissions also followed the overall substance-related disorder association patterns (e.g., an increase in temperature from daily minimum ($-30.1\,°C$ ($-22.2\,°F$)) to 75th percentile ($18.8\,°C$ ($65.8\,°F$)) across 0–6 lag days was associated with a cumulative 37.6% (95%CI, $-20.3$%–55.0%) increase in hospital visit rates). For opioid-related admissions, there was an increase up to the mean temperature with a decrease above that. Sedative-related admissions indicated no overall discernible change across the temperature range.

**Association of temperature with hospital visits by location.** For alcohol-related disorders, there was no discernible difference in increases of hospital visit rates at warmer temperatures outside NYC compared with NYC (e.g., an increase in temperature from the daily minimum ($-30.1\,°C$ ($-22.2\,°F$)) to 75th percentile ($18.8\,°C$ ($65.8\,°F$)) was associated with a cumulative 25.7% (95% CI, 14.0–37.4%) increase for outside NYC compared with a cumulative 24.6% (95% CI, 8.7–40.5%) increase for NYC) (Fig. 4). For substance-related disorders, there was a larger increase in hospital visit rates at warmer temperatures outside NYC compared with NYC (e.g., an increase in temperature daily minimum ($-30.1\,°C$ ($-22.2\,°F$)) to 75th percentile ($18.8\,°C$ ($65.8\,°F$)) was associated with a cumulative 49.8% (95% CI, 37.7–61.9%) increase for outside NYC compared with a cumulative 14.3% (95% CI, $-0.4$–29.0%) increase for NYC). For sub-

causes of substance-related disorders, results were inconclusive, though point estimates for outside NYC were consistently higher (Supplementary Fig. 3). Other sub-analyses (by females vs. males, age group, social vulnerability) demonstrated consistent evidence of an association (Supplementary Figs. 4–9). Though Supplementary Fig. 6 indicates that the least socially vulnerable tertile may have a higher-in-magnitude association than that of the more vulnerable tertiles for the 90th and 99th percentile temperature values, we cannot conclusively state that the associations are higher or lower given the overlapping 95% confidence intervals.

**Secondary analyses.** Results of analyses only including temperatures from the day of and the day before (0–1 days) for associations of temperature with total hospital visits for causes are found in Supplementary Figs. 10 and 11. The association of alcohol-related disorders at higher temperatures was potentially attenuated and with lower uncertainty, though still a positive association (Supplementary Fig. 10) (e.g., an increase in temperature from the daily minimum ($-30.1\,°C$ ($-22.2\,°F$)) to 75th percentile ($18.8\,°C$ ($65.8\,°F$)) across 2 lag days was associated with a cumulative 19.0% (95% CI, 12.2–25.8%) increase). The association of substance-related disorders at higher temperatures at higher temperatures was also potentially attenuated with lower uncertainty) (e.g., an increase in temperature from the daily minimum ($-30.1\,°C$ ($-22.2\,°F$)) to 75th percentile ($18.8\,°C$ ($65.8\,°F$)) across 2 lag days was associated with a cumulative 24.0% (95% CI, 17.2–30.7%) increase) (Supplementary Fig. 10) Overall conclusions were the same as the main analyses.

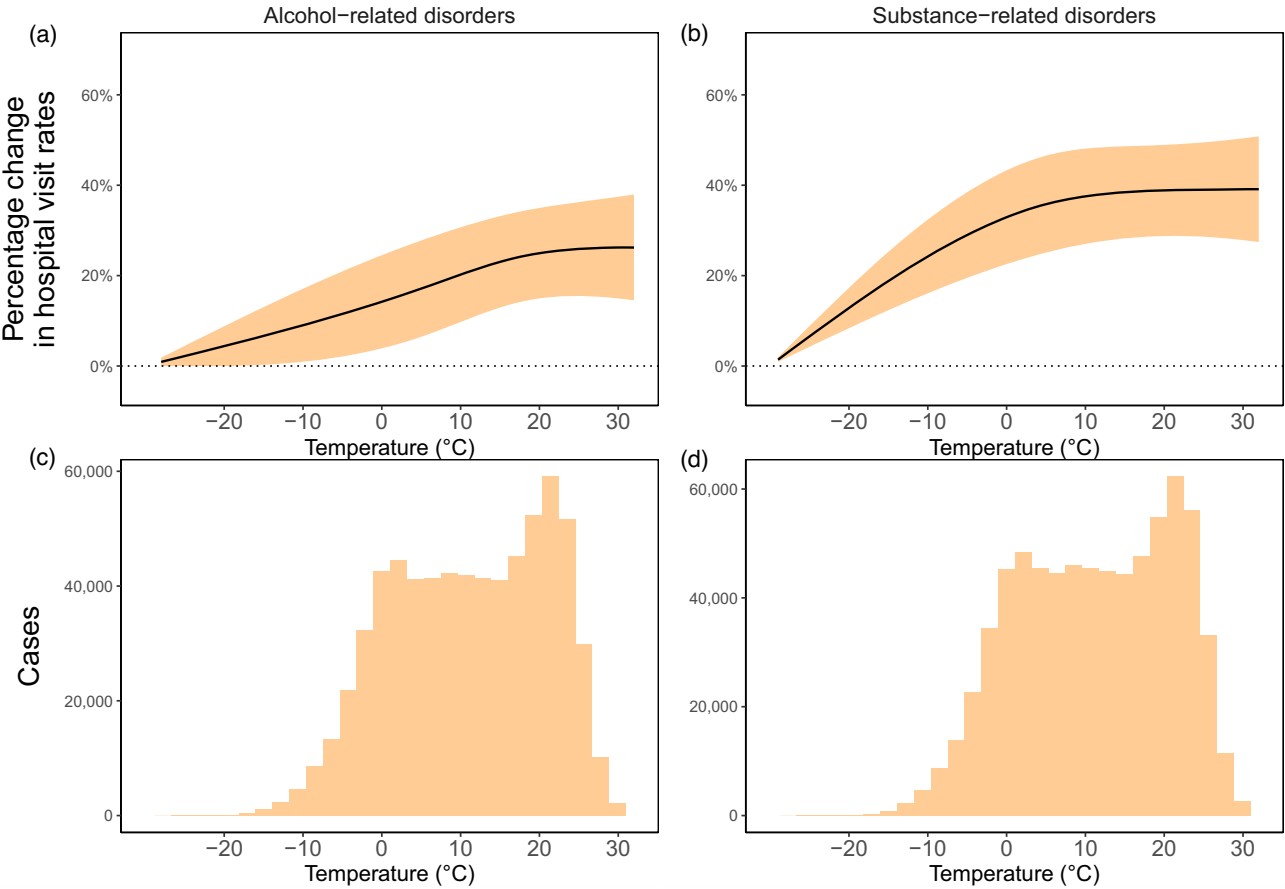

**Fig. 3 Exposure-response curves of cumulative percentage change in hospital visit rates relative to minimal temperature (−30.1 °C (−22.18 °F)) and histograms of temperature records for cases.** Percentage changes represent the scenario where each of the lag days (0 to 6 days before) at the quoted temperature before hospital visit. Exposure-response curves for **a** alcohol- (*n* = 671,625) and **b** substance-related (*n* = 721,469) disorder hospital visits, and histograms of temperature records for cases of **c** alcohol- (*n* = 671,625) and **d** substance-related (*n* = 721,469) disorder hospital visits. Black lines show the point estimates and orange ribbons represent 95% confidence intervals.

**Sensitivity analyses**. For relative humidity sensitivity analyses (Supplementary Fig. 12), there was a correlation of R = 0.99 and a slope of 0.99 (95% CI, 0.96–1.01) between estimates of associations with (main) and without (sensitivity) relative humidity in the model.

## Discussion
In NYS from 1995 to 2014, an increase in temperature 0–6 days before hospital visits was associated with higher hospital visit rates for both alcohol- and substance-related disorders up to a threshold, above which no increases were discernible.

**Plausibility of results**. That temperature influences hospital visits from both alcohol- and substance-related disorders, although not previously quantified, is plausible. Changes in alcohol- and substance-related hospital visits may result from changes in temperature for many behavioral or psychological reasons. Overall decreases in hospital visit rates below average temperatures may also be driven by lower enthusiasm to visit the hospital, as it may, for example, seem more dangerous in particularly cold or inclement weather[25], especially while under the influence of a psychoactive substance. Higher hospital visits in higher temperatures for alcohol-related disorders may potentially be driven by more time outdoors performing riskier activities, consuming more substances in more pleasant outdoor weather, more perspiration causing greater dehydration, or driving while under the influence[8]. The observed temperature thresholds may be because

once outdoor temperatures are sufficiently comfortable, further temperature increases may not increase outdoor activity. There was no clear evidence of increases in substance-related disorder hospital visits for temperatures higher than the mean temperature. Nevertheless, for cocaine, there was a potential increase for higher temperatures, which may be driven by the consumption of alcohol and increased sweating, increasing the risk of cardiovascular and respiratory issues developing. Those who regularly take opioids have found that their efficacy is reduced in warmer weather, and may potentially take higher doses on warmer days[9]. Hospital visits for alcohol- and substance-related disorders are usually emergency visits[39], which would indicate that the association between temperature and hospital visits is driven largely by those who make unscheduled visits to the hospital.

**Alcohol and substance use in the United States over recent decades**. These results should be taken in the context of the past few decades that New York State and the United States have experienced. The opioid epidemic in the United States during the past few decades has resulted in a large increase in usage and dependency on prescription and illicitly-sourced opioids[40]. Drug overdose deaths in the United States have increased more than five times since the end of the 20th century[41]. Alcohol-attributable deaths during 2015–2019 have been estimated to represent 1 in 8 deaths among adults aged 20 to 64 years[3]. The positive association of short-term temperature exposure and alcohol- and substance-related disorder hospital visits could further exacerbate

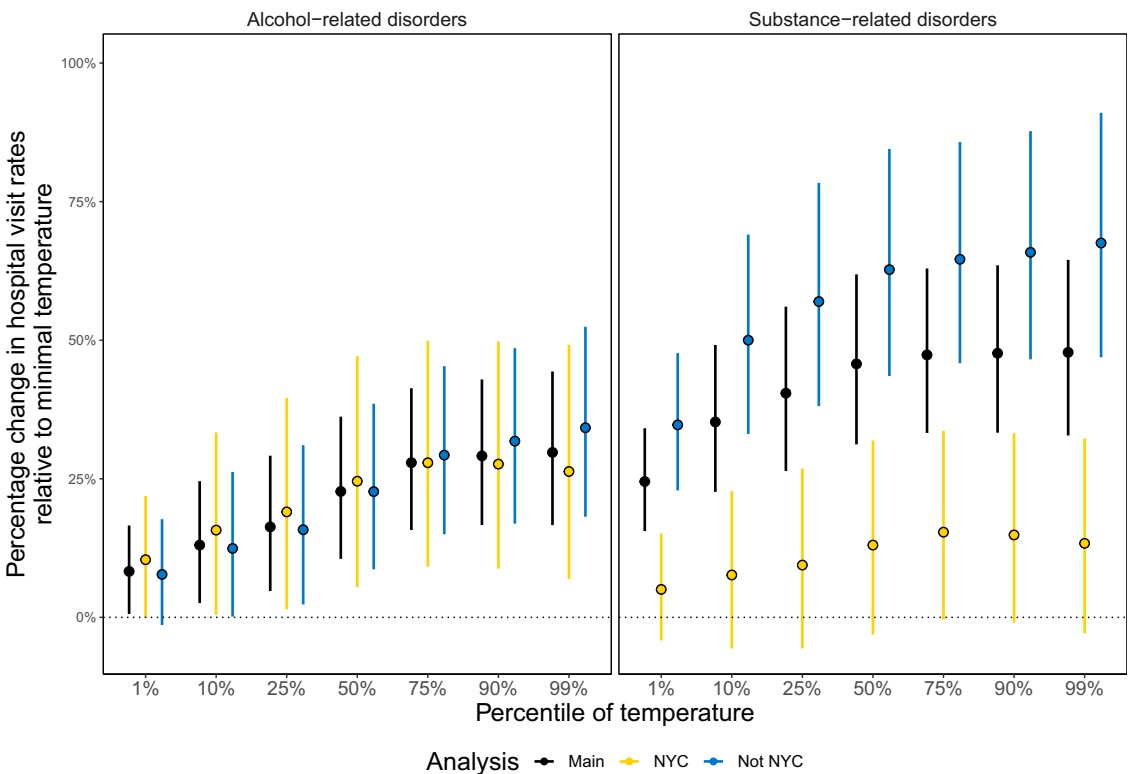

**Fig. 4 Percentage change in hospital visit rates by selected percentiles of temperature relative to minimal temperature (−30.1 °C (−22.18 °F)).**
Percentage changes represent the scenario where each of the lag days (0 to 6 days before) at the quoted temperature before the hospital visit. Percentile plots for alcohol- *n* = 671,625) and substance-related (*n* = 721,469) disorder hospital visits by location in New York State. Points show the point estimates and whiskers represent 95% confidence intervals.

negative alcohol- and substance-related outcomes in the United States with rising temperatures under climate change.

**Strengths and limitations**. Leveraging complete hospital visit data from 671,625 alcohol- and 721,469 substance-related disorder hospital visits over 20 years and a comprehensive record of ZIP Code-level daily temperatures and relative humidity, this study is the first, to our knowledge, comprehensive investigation of the association between temperature and alcohol- and substance-related hospital visits.

The study has several limitations. First, a potential limitation is outcome misclassification, as it is likely that the most severe cases of alcohol- and substance-related disorders resulted in deaths before hospital visit was possible. Future work should attempt to link cases of deaths with hospital visit records to create a fuller picture of patients' medical history. Second, exposure misclassification is inevitable, e.g., if those who were hospitalized were located at a different ZIP Code than their residential ZIP Code. However, it is not very likely that a large proportion of the cases would be away from their residential ZIP Code for the week—the exposure window we examined—prior to a hospital visit. Exposure misclassification, therefore, is likely non-differential as it is not expected to be correlated with the outcomes assessed, potentially biasing towards the null[42]. Third, the estimated associations may have been susceptible to confounding bias. By matching using the time-stratified case-crossover structure, where cases are matched to themselves during periods where they were not hospitalized, this design controls for factors varying across individuals, as well as day of the week, month, and season, but the possibility of residual confounding by unknown or unmeasured factors which vary over the time scale of a few weeks cannot be

ruled out. Any such variable, however, would have to covary with both hospital visit rates and temperature in ZIP Codes and be independent of the variables included in analyses to induce residual or unmeasured confounding. Fourth, this study was focused on New York State, though temperature is a pervasive exposure and the association with alcohol- and substance-related disorders should be further explored in locations with different communities and climates. Fifth, the consequences of these findings in the context of a changing climate are unclear. Adaptation may play a key role in mitigating the worst impacts of climate change on health[43]. However, there are limits to the adaptive capacity of humans, and these results should be further explored in the context of adaptive capacity. Nevertheless, these results further indicate that public health practitioners preparing for climate change should consider outcomes such as substance use disorder or mental health events that are not usually linked to temperature. Sixth, this study focused on each individual alcohol- and substance-related disorder per analysis. Further work should examine the role of co-morbidities, as well as whether existing health conditions are exacerbated by alcohol and/or substance use combined with rising temperatures. Seventh, due to data access restrictions, this study focused on hospital visits in total, not just first-time visits. Further work should examine the effect modification between those who make their first alcohol- or substance-related disorder hospital visits vs. re-hospitalizations. Eighth, we were not able to distinguish between those patients with permanent addresses or those without. Further work should focus on the unhoused population of New York State, who are potentially particularly vulnerable to the health impacts of rising temperatures. Ninth, we did not examine co-morbidities, though there may be particular groups with existing co-morbidities that

modify their alcohol- and substance-related vulnerability to heat stress.

Our work highlights how hospital visits from alcohol- and substance-related disorders are currently susceptible to elevated temperatures and could also be modified by rising temperatures resulting from climate change, unless countered by social infrastructure and health system interventions that mitigate these impacts. Public health interventions that broadly target alcohol and substance disorders in warmer weather—for example, targeted messaging on the risks of their consumption during warmer weather—should be a public health priority.

## Data availability

NLDAS-2 temperature, specific humidity, and pressure data are downloadable from https://disc.sci.gsfc.nasa.gov/datasets?keywords=NLDAS. SPARCS hospital records can be requested through the submission of a proposal to the New York State Department of Health (https://www.health.ny.gov/statistics/sparcs/). All data populating Figures and Tables throughout the analysis and visualization presented in this manuscript are publicly available via https://github.com/rmp15/temperature_alcohol_substance_communications_medicine_2023 and archived in Zenodo[44].

## Code availability

All code for analysis, results from analysis, and visualization presented in this manuscript is publicly available via https://github.com/rmp15/temperature_alcohol_substance_communications_medicine_2023 and archived in Zenodo[44].

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

## Acknowledgements

Robbie M Parks was supported by the NIEHS K99 ES033742, NIEHS R00 ES033742 and the Earth Institute post-doctoral research fellowship at Columbia University. Funding was also provided by the National Institute of Environmental Health Sciences (NIEHS) grants R01 ES030616, R01 ES028805, P30 ES009089, T32 ES023770, and R21 MD012451. The funders had no role in study design, data collection and analysis, the decision to publish, or preparation of the manuscript.

## Author contributions

R.M.P. and M.A.K. contributed to the study design. R.M.P., S.T.R., V.D., A.K.B., C.L.H., and M.A.K. acquired, analyzed, and interpreted data. R.M.P. and M.A.K. obtained funding. R.M.P. and M.A.K. conducted analysis and prepared results. R.M.P. and M.A.K. wrote the first draft of the paper, and S.T.R., V.D., F.D., A.K.B., and C.L.H. contributed to critical revisions.

## Competing interests

The authors declare no competing interests.
