## [Peer Review File · Communications Medicine]

Reviewers' comments:

Reviewer #1 (Remarks to the Author):

This study addresses the gap in knowledge regarding how temperature increases are associated with hospitalizations from alcohol- and substance-related disorders (i.e., cannabis, cocaine, opioid, sedative disorders). They used a case-crossover design to control for season and confounding bias, implemented with distributed lag non-linear temperature terms (0-6 days) to estimate associations between daily ZIP Code-level temperature and alcohol- and substance-related disorder hospitalization rates in New York State during 1995–2014. The paper would be improved with the following revisions.

Introduction:

1. The first sentence of the paper does not connect well to the study aims.
2. The authors have structured the aims to focus on hospitalizations for people with SUD/AUD diagnoses – which means that the population of interest is likely those with previously identified and diagnosed SUD/AUD disorders (some may have had their first diagnosis during the heat event, but this wasn't looked at). Therefore, the introduction should focus on people in the US with substance use disorders. The introduction can be enhanced in this area.
3. The authors should briefly describe what is known to date on this topic in the background – some literature is reported later in the discussion that could have been briefly introduced in the background.
4. The authors should receive the recent JAMA Psych paper by Nori-Sarma et al regarding the association between ambient heat and risk of ED visits for Mental Health Among US Adults, 2010 to 2019 – which included an outcome specific to SUD. doi:10.1001/jamapsychiatry.2021.4369

Methods:

1. I am unclear why the methods section is at the end of the paper after the references. I read the paper thinking it was missing and therefore had a hard time reviewing the paper with the required information. Then I saw it at the end. I looked at the author instructions for the journal and didn't see anything about putting the methods after the references.
2. The outcome is described as substance-related disorder hospitalizations, yet the authors report that the outcome could have been inpatient or outpatient. Hospitalizations are considered inpatient. Should language be modified? More information is needed.
3. By using a study design that starts with hospitalizations during periods of high temperature the authors are not able to examine the association of extreme heat with hospitalizations for substance use related diagnoses more broadly. They do not necessarily capture all extreme temperature days and then look for likelihood of hospitalization for SUD related causes. The authors should describe why they used the analytic approach that they took and make sure the language used in the title, abstract and text accurately describes what their results tell us.

Results:

4. Are the authors able to report results in both Celsius and Fahrenheit? This study uses data from the US where Fahrenheit is more broadly reported and understood.

Discussion:

5. The study covers a long period of time in the US (1995-2014) in which there were many important changes happening related to the opioid epidemic (driven by both prescription opioids and then illicit opioids) as well as an increase in alcohol-related morbidity and mortality. The discussion would be

improved by incorporating some of this context into their discussion.

Reviewer #2 (Remarks to the Author):

Parks and colleagues investigated the association between air temperature and alcohol- and substance-related disorder hospitalizations in New York State. Using a time-stratified case-crossover design, this study found that rising temperatures were associated with higher hospitalization rates for these two disorders. Air temperature has been linked to a variety of health outcomes, but its relationship with alcohol- and substance-related disorders is understudied. This study contributes a novel piece of evidence to the health effects of air temperature. However, there are several issues that should be addressed before getting published in *Communications Medicine*.

Major Comments

1. One of my major concerns is about the exposure assessment for homeless people, as the outcome of interest is alcohol- and substance-related disorder, and a high percentage of homeless people do struggle with substance abuse. The authors linked temperature data to each patient based on the patient's residential ZIP Code, but what if there is no fixed residence?
2. Since this study focuses on short-term acute effects, it is important to ensure that the date of hospitalization reflects the date when the symptoms onset or exacerbated. Does the SPARCS dataset contain the information on whether a hospital visit is a walk-in or by appointment? It is recommended to exclude those by appointment because, for those cases, the temperature on the day of hospitalization and the preceding days should be considered independent of their diseases.
3. The near-linear relationship on the cold side needs more explanation (Figure 3). The authors explained this finding by lower enthusiasm to visit hospital in cold weather (lines 182-184), but if this is the case, there should be a threshold on the cold side too, below which the unwillingness to visit hospital should remain the same.
4. In the stratified analysis by social vulnerability, the effect estimates are higher in the least vulnerable group, compared to the most vulnerable group (Supplementary Figure 6). This finding is unexpected because previous literature usually found higher risks in more vulnerable populations. More discussion on this finding is recommended.
5. Does the SPARCS dataset has a patient ID, which enables the authors to detect re-hospitalization? If so, a stratified analysis by first-time admission vs. re-hospitalization would be helpful to investigate the potential differences in the effects of temperature on these two types of hospitalization. In addition, if re-hospitalizations are common in the datasets, mixed effects models could be considered.

Minor Comments

1. Figure 3: Adding a layer of temperature distribution in this figure would be helpful for readers to know which part of the estimated curve is more reliable. In addition, instead of using the mean temperature as the reference temperature, how about using the median value, or the temperature corresponding to the lowest risk? How will different centers influence the estimated curve?
2. Based on the estimated curve, the number of temperature-attributable hospitalizations can also be calculated to show the burden of alcohol- and substance-related disorder hospitalizations contributed by air temperature.
3. Does the effects of temperature modified by co-morbidities? It worth a secondary analysis if information on co-morbidity is available.
4. Line 393-394: More sensitivity analyses on model parameters are needed to check the robustness of the results. For example, different degrees of freedom (df), lag days, and spline types.
5. Besides the studies that were already mentioned in the manuscript, is there any other evidence that supports a relationship between temperature and alcohol- and substance-related disorders? In addition to human studies, are there any related animal studies?
6. Supplementary Figures 3-9: The limits of the y-axis need to be reset so the central estimate and the error bars can be fully displayed (e.g., Figure S3, Sedatives – NYC – 1%).

Reviewer #3 (Remarks to the Author):

Thank you very much for letting me review this interesting work. The authors present a novel investigation on the association between ambient temperature and hospitalizations due to substance abuse and alcohol. I believe it is a timely contribution with relevant findings for public health. The topic of mental health in the context of climate change is considered today a hot topic, given the recent evidence on the direct and indirect connection between climate hazards and the consequences for mental health patients. The research presented is new, and the method used here is robust. I would only suggest the authors to clarify a few aspects on the interpretation of the findings and method.

Abstract

- "which controls for season and confounding bias" - this is not accurate, case-crossover designs inherently control for temporal trends (both seasonal and long term trends) and time-invariant confounders or those time-varying factors that change at a longer time scale than the association of interest.
- I would suggest the authors to avoid repeating the results in the conclusion and invest the space into providing a broader context - for example, the public health implications in relation to two relevant threats - climate change and poor mental health connected with the abuse of substances in the US.

Intro

- In the introduction, I would suggest the authors to highlight the synergistic effect between increasing variability due to climate change, and the rise of mental health problems due to accelerated socioeconomic trends and poor healthcare system that lead to the abuse of substances in (young) adults.
- In the aims, the authors state "how short term changes in exposure to daily temperature was associated with hospitalizations..." in this way, the readers would consider that the exposure variable is the "change in temperature" but it is actually not. The method here provides RRs that are measures of change in hospitalization risk based on different levels of temperature (18C vs. 10C).

Results

- In the descriptive analysis, I would suggest the authors to provide (age-standardized) hospitalization rates by ZIP or across time (to complement the absolute number of hospitalizations). In the way how it is reported now, one cannot disentangle whether the spatial and temporal patterns are just driven by changes in the demographic structure of the population.
- Related to the comment above, I would avoid using "daily" increases of temperature - it would be prone to misunderstandings, since it seems that it refers to changes in temperature from one day to the other (e.g. 107 a daily increase from the daily average to the 90th percentile).
- Not sure whether the use of "increase in hospitalization rates" since in this context we are talking about increase in risk (e.g., line 107)
- The authors mention in line 111 that " we did not find evidence of an association " with an estimate 1.2% (95%CI, -0.1%–2.4%) - it is true that the association estimate is smaller compared to the other estimate, and non-statistically significant. I would be more incline to interpret this finding as there seems to be an indication of an association although smaller and non-statistically significant.
- Related to the previous comment, I don't agree with the authors of using the mean temperature value as reference - why not using the 10th percentile or the minimum value? so most of the temperature range is above that value and it is easier to interpret. I would suggest the authors to first describe the shape in both cases - close to linear with a kind of plateau at very high temperatures. This shape has been found in other studies on mental health and temperature (suicides - Kin et al <https://doi.org/10.1289/EHP4898>. ; Barr et al 2022 doi:10.4414/SMW.2022.w30115).

We thank the Editors and Reviewers for their thoughtful and constructive suggestions. We have revised the manuscript in response to the Reviewers' comments, as detailed below.

All page/line/reference numbers refer to the tracked revised manuscript.

Reviewers' comments:

Reviewer #1 (Remarks to the Author):

This study addresses the gap in knowledge regarding how temperature increases are associated with hospitalizations from alcohol- and substance-related disorders (i.e., cannabis, cocaine, opioid, sedative disorders). They used a case-crossover design to control for season and confounding bias, implemented with distributed lag non-linear temperature terms (0-6 days) to estimate associations between daily ZIP Code-level temperature and alcohol- and substance-related disorder hospitalization rates in New York State during 1995–2014. The paper would be improved with the following revisions.

We thank the Reviewer for the thoughtful and constructive suggestions. We have responded point-by-point to the Reviewer's questions and comments below.

Introduction:

1. The first sentence of the paper does not connect well to the study aims.

We have removed this sentence from the revised manuscript.

2. The authors have structured the aims to focus on hospitalizations for people with SUD/AUD diagnoses – which means that the population of interest is likely those with previously identified and diagnosed SUD/AUD disorders (some may have had their first diagnosis during the heat event, but this wasn't looked at). Therefore, the introduction should focus on people in the US with substance use disorders. The introduction can be enhanced in this area.

In the Introduction of the revised manuscript, we have added a focus of those in the United States who are living with substance use disorders (P. 3, Lines 64-68):

However, a substantial minority of individuals suffer from related disorders, defined as when recurrent use of alcohol or other substances causes clinically significant impairment,² and require intervention, including inpatient treatment. Many individuals requiring intervention are regular substance users.¹ One in eight deaths in working-age Americans are estimated to be associated with excessive alcohol use.³

3. The authors should briefly describe what is known to date on this topic in the background – some literature is reported later in the discussion that could have been briefly introduced in the background.

We have substantially added to the description of background knowledge in the existing literature in the revised manuscript, including what was previously in the discussion (PP. 4-5, Lines 90-121):

Limited previous studies of alcohol- and substance-related disorders in relation to temperature exist. A near-linear association was found between alcohol or drug poisonings discovered during unhoused rescue missions and temperature in Hamburg, Germany,¹³ while another study of alcohol use disorders in Paris, France found a correlation of 0.55 between weekly alcohol use disorders and mean temperature.¹⁴ However, neither study adequately controlled for confounding bias, including season. Heatwaves in Hanoi, Vietnam, were associated with increases in admissions from mental disorders in general, but not from psychoactive substance use.¹⁵ Substance abuse-related mental illness emergency hospital visits in Toronto, Canada were positively associated with higher temperatures.¹⁶ A study leveraging information on Optum hospital visits, a selected sub-set of total United States hospital visits based on insurance records, identified a positive association with temperature and substance-related disorders.¹⁷ Studies of rats have found that high ambient temperatures impact the acquisition of 3,4-methylenedioxymethamphetamine (MDMA) and methamphetamine, as well as dopamine self-administration.¹⁸⁻²¹ Nevertheless, there remains an overall knowledge gap in consistently and comprehensively quantifying how temperature is associated with alcohol- and substance-related hospital visits.

4. The authors should receive the recent JAMA Psych paper by Nori-Sarma et al regarding the association between ambient heat and risk of ED visits for Mental Health Among US Adults, 2010 to 2019 – which included an outcome specific to SUD. doi:10.1001/jamapsychiatry.2021.4369

We now include the Nori-Sarma et al. study in the revised manuscript (P. 4, Lines 99-101):

A study leveraging information on Optum hospital visits, a selected sub-set of total United States hospital visits based on insurance records, identified a positive association with temperature and substance-related disorders.¹⁷

Methods:

1. I am unclear why the methods section is at the end of the paper after the references. I read the paper thinking it was missing and therefore had a hard time reviewing the paper with the required information. Then I saw it at the end. I looked at the author instructions for the journal and didn't see anything about putting the methods after the references.

We have moved the Methods section to immediately after the Introduction (PP. 5-10, Lines 129-260).

2. The outcome is described as substance-related disorder hospitalizations, yet the authors report that the outcome could have been inpatient or outpatient. Hospitalizations are considered inpatient. Should language be modified? More information is needed.

We have modified the language throughout the revised manuscript to describe events as hospital visits, e.g., (P. 5, Lines 123-127):

The aim of this study was to evaluate (a) how ~~short term changes in exposure to~~ daily temperature was associated with ~~hospitalizations~~ hospital visits due to alcohol- or substance-related disorders (including alcohol, cannabis, cocaine, opioids and sedatives), and (b) how this association varied by location, age group, sex, and social vulnerability, using daily ZIP Code-level ~~hospitalization~~ hospital visit data obtained from hospitals in NYS, the fourth largest state by population in the United States.⁴⁰²²

3. By using a study design that starts with hospitalizations during periods of high temperature the authors are not able to examine the association of extreme heat with hospitalizations for substance use related diagnoses more broadly. They do not necessarily capture all extreme temperature days and then look for likelihood of hospitalization for SUD related causes. The authors should describe why they used the analytic approach that they took and make sure the language used in the title, abstract and text accurately describes what their results tell us.

Our study design, implementing a case-crossover design with distributed lag non-linear temperature terms, utilizes the entire temperature range available in the data, and not simply the period of high temperatures. As hospital records are available for every day during our study period, the temperature records will also be complete.

As can be seen in Figure 3, for example, copied below for reference, the exposure-response curve is generated for the entire available temperature range. This means that we do not only use hospital visit information during periods of high temperature, but every single alcohol-related or substance-related hospital visit that occurred in New York State during our study period.

Figure 3. Exposure-response curves (top) of cumulative percentage change in hospitalization hospital visit rates relative to average minimal temperature (10(-30.1°C) (-22.18°F)) for alcohol- and substance-related disorder hospitalizations hospital visits, were each of the lag days (0 to 6 days before) at the quoted temperature before hospitalization hospital visit. Black lines show the point estimates and orange ribbons represent 95% confidence intervals. Histograms (bottom) of temperature records for cases.

We have clarified this in the Methods of the revised manuscript (PP. 7-8, Lines 210-214):

A time-stratified case-crossover design was used, commonly used for analyzing associations with short-term exposures.^{33,34} In this design, temperature of the day of hospital visit and relevant preceding days (case period) are compared with the temperature of sets of days where the hospital visit did not occur (control periods). This study design utilizes every single hospital visit, not only those during periods of high temperatures.

Results:

4. Are the authors able to report results in both Celsius and Fahrenheit? This study uses data from the US where Fahrenheit is more broadly reported and understood.

We have added Fahrenheit as a reference in the relevant Figures (example in the revised Figure 3 caption above), and throughout the revised manuscript as a reference for the American audience, e.g., (P. 11, Lines 310-314):

For alcohol-related disorder ~~hospitalizations, a daily~~hospital visits, an increase in temperature from the period ~~daily average (10~~minimum (-30.1°C) (-22.2°F)) to the 75th percentile (18.48°C) (65.8°F)) across 0-6 lag days was associated with a cumulative ~~4.3%–24.6%~~ (95%CI, ~~3.0%–5.5~~14.6%–34.6%) increase in ~~daily hospitalization~~hospital visit rates, and ~~a daily~~an increase from the ~~daily~~average to the 90th percentile (22.37°C) (72.9°F)) was associated with a cumulative ~~5.3~~25.6% (95%CI, ~~3.5%–15.4%–35.7~~0%) increase (Figure 3).

Discussion:

5. The study covers a long period of time in the US (1995-2014) in which there were many important changes happening related to the opioid epidemic (driven by both prescription opioids and then illicit opioids) as well as an increase in alcohol-related morbidity and mortality. The discussion would be improved by incorporating some of this context into their discussion.

We have added further context in Introduction of the revised manuscript P. 3, Lines 68-70):

In recent decades, there has been an increasing trend of heavy episodic drinking and alcohol-related morbidity and mortality in the United States, particularly in middle-aged to older adults.⁴

We have also added a section in the discussion of the revised manuscript which discusses the wider context of our results in the United States' recent history of alcohol and substance use (PP. 15-16, Lines 592-604):

These results should be taken in the context of the past few decades that New York State and the United States have experienced. The opioid epidemic in the United States during the past few decades has resulted in a large increase in usage and dependency on prescription and illicitly-sourced opioids.⁴¹ Drug overdose deaths in the United States have increased more than five times since the end of the 20th century.⁴² Alcohol-attributable deaths during 2015-2019 have been estimated to represent 1 in 8 deaths among adults aged 20 to 64 years.³ The positive association of short-term temperature exposure and alcohol- and substance-related disorder hospital visits could further exacerbate negative alcohol- and substance-related outcomes in the United States with rising temperatures under climate change.

Reviewer #2 (Remarks to the Author):

Parks and colleagues investigated the association between air temperature and alcohol- and substance-related disorder hospitalizations in New York State. Using a time-stratified case-crossover design, this study found that rising temperatures were associated with higher hospitalization rates for these two disorders. Air temperature has been linked to a variety of health outcomes, but its relationship with alcohol- and substance-related disorders is understudied. This study contributes a novel piece of evidence to the health effects of air temperature. However, there are several issues that should be addressed before getting published in Communications Medicine.

We thank the Reviewer for the thoughtful and constructive suggestions. We have responded point-by-point to the Reviewer's questions and comments below.

Major Comments

1. One of my major concerns is about the exposure assessment for homeless people, as the outcome of interest is alcohol- and substance-related disorder, and a high percentage of homeless people do struggle with substance abuse. The authors linked temperature data to each patient based on the patient's residential ZIP Code, but what if there is no fixed residence?

In New York State, most of the unhoused (homeless) community lives in New York City (67,150 in NYC of 91,271 in NYS according to <https://worldpopulationreview.com/state-rankings/homeless-population-by-state>), across which temperature gradients are relatively small, i.e., on the scale of a city. This means that exposure misclassification for an unhoused person visiting a hospital for an alcohol- or substance-related disorder (and recorded in the SPARCS dataset we used for this analysis) would be small.

We do not currently have information related to whether a person is unhoused at the time of hospital visit. We have added this as a limitation for examination in future work in the Discussion of the revised manuscript (PP. 17-18, Lines 653-659):

Eighth, we were not able to distinguish between those patients with permanent addresses or those without. Further work should focus on the unhoused population of New York State, who are potentially particularly vulnerable to the health impacts of rising temperatures.

2. Since this study focuses on short-term acute effects, it is important to ensure that the date of hospitalization reflects the date when the symptoms onset or exacerbated. Does the SPARCS dataset contain the information on whether a hospital visit is a walk-in or by appointment? It is recommended to exclude those by appointment because, for those cases, the temperature on the day of hospitalization and the preceding days should be considered independent of their diseases.

The SPARCS dataset does not include whether a hospital visit was a walk-in or an appointment. However, a large majority of alcohol and substance-related disorder hospital visits are typically

emergency visits. We have added this to the Discussion of the revised manuscript (P. 15, Lines 587-589):

Hospital visits for alcohol- and substance-related disorders are usually emergency visits,⁴⁰ which would indicate that the association between temperature and hospital visits is driven largely by those who make unscheduled visits to hospital.

3. The near-linear relationship on the cold side needs more explanation (Figure 3). The authors explained this finding by lower enthusiasm to visit hospital in cold weather (lines 182-184), but if this is the case, there should be a threshold on the cold side too, below which the unwillingness to visit hospital should remain the same.

We apologize, but we are unsure what the Reviewer means by this comment. However, in response to other comments by Reviewers, we have changed the reference temperature for Figure 3 to the minimal temperature recorded. Figure 3 is copied below for reference:

Figure 3. Exposure-response curves (top) of cumulative percentage change in hospitalization hospital visit rates relative to average minimal temperature (10(-30.1°C) (-22.18°F)) for alcohol- and substance-related disorder hospitalizations hospital visits, were each of the lag days (0 to 6 days before) at the quoted temperature before hospitalization hospital visit. Black lines show the point estimates and orange ribbons represent 95% confidence intervals. Histograms (bottom) of temperature records for cases.

If we can further clarify, we would be happy to.

4. In the stratified analysis by social vulnerability, the effect estimates are higher in the least vulnerable group, compared to the most vulnerable group (Supplementary Figure 6). This finding is unexpected because previous literature usually found higher risks in more vulnerable populations. More discussion on this finding is recommended.

Though Supplementary Figure 6 indicates that SVI tertile 1 has a point estimate which is higher than that of SVI 2 and SVI 3 for the 90th and 99th percentile temperature values, the overlapping 95% confidence intervals mean that it cannot be conclusively stated that the associations are higher or lower. We have explicitly mentioned this in the Results of the revised manuscript (P. 13, Lines 470-474):

Though Supplementary Figure 6 indicates that the least socially vulnerable tertile may have a higher-in-magnitude association than that of the more vulnerable tertiles for the 90th and 99th percentile temperature values, we cannot conclusively state that the associations are higher or lower given the overlapping 95% confidence intervals.

5. Does the SPARCS dataset has a patient ID, which enables the authors to detect re-hospitalization? If so, a stratified analysis by first-time admission vs. re-hospitalization would be helpful to investigate the potential differences in the effects of temperature on these two types of hospitalization. In addition, if re-hospitalizations are common in the datasets, mixed effects models could be considered.

Our revised paper includes analysis by in-patient and out-patient visits. Our access to the SPARCS dataset unfortunately does not include patient ID, and applying and waiting for access to data with patient ID may delay the paper by several years.

Nevertheless, we certainly recognize that a first-time admission vs. re-hospitalization comparison would be a useful analysis to carry out in the future, and have acknowledged that in the limitations and future work (P. 17, Lines 649-652):

Seventh, due to data access restrictions, this study focused on hospital visits in total, not just first-time visits. Further work should examine the effect modification between those who make their first alcohol- or substance-related disorder hospital visits vs. re-hospitalizations.

Minor Comments

1. Figure 3: Adding a layer of temperature distribution in this figure would be helpful for readers to know which part of the estimated curve is more reliable. In addition, instead of using the mean temperature as the reference temperature, how about using the median value, or the temperature corresponding to the lowest risk? How will different centers influence the estimated curve?

We have now added a panel with temperature distribution to the figure to help readers understand how the different parts of the curve are informed by data.

We now quote the risk of changes from the minimal temperature in the range (-30.1°C (-22.18°F); lowest risk temperature), e.g., (P. 11, Lines 310-314):

For alcohol-related disorder ~~hospitalizations, a daily hospital visits, an~~ increase in temperature from the period ~~daily average (10 minimum (-30.1°C) (-22.2°F))~~ to the 75th percentile (18.48°C) (65.8°F) across 0-6 lag days was associated with a cumulative ~~4.3%–24.6%~~ (95%CI, ~~3.0%–5.5~~14.6%–34.6%) increase in ~~daily hospitalization~~hospital visit rates, and ~~a daily an~~ increase from the ~~daily average to the 90th percentile (22.37°C) (72.9°F)~~ was associated with a cumulative ~~5.3~~25.6% (95%CI, ~~3.5%–15.4%–35.7~~0%) increase (Figure 3).

The revised Figure 3 can be seen below for reference:

Figure 3. Exposure-response ~~curve~~curves (top) of cumulative percentage change in ~~hospitalization~~hospital visit rates relative to ~~average~~minimal temperature (~~10(-30.1°C) (-22.18°F)~~) for alcohol- and substance-related disorder ~~hospitalizations~~hospital visits, were each of the lag days (0 to 6 days before) at the quoted temperature before ~~hospitalization~~hospital visit. Black lines show the point estimates and orange ribbons represent 95% confidence intervals. Histograms (bottom) of temperature records for cases.

2. Based on the estimated curve, the number of temperature-attributable hospitalizations

can also be calculated to show the burden of alcohol- and substance-related disorder hospitalizations contributed by air temperature.

We agree with the Reviewer that this is an interesting suggestion, albeit outside the scope of our study. Here we utilize only cases of hospital visits for our case crossover analysis; adding population information and estimating baseline rates would add an additional layer of complexity to the results, which can potentially become too large and disconnected. Since the Reviewer is not necessarily requesting this, we have not done this.

3. Does the effects of temperature modified by co-morbidities? It worth a secondary analysis if information on co-morbidity is available.

We thank the Reviewer for this suggestion. We think this will be an interesting future study. However, it is currently unclear on which groupings would be aetiologically plausible to examine too. We would thus prefer to not add too many additional analyses, to avoid multiple comparisons issues. Importantly, we unfortunately do not have a comprehensive history of co-morbidities, just the additional diagnostic codes in the substance- or alcohol-abuse hospitalization. Therefore, any analysis using these data could potentially include large misclassification in the modifiers (comorbidities) examined.

We have mentioned this in the Strengths and limitations of the revised manuscript (P. 17, Lines 646-649):

Sixth, this study was focused on each individual alcohol- and substance-related disorder per analysis. Further work should examine eases the role of co-morbidities, as well as whether existing health conditions are exacerbated by alcohol and/or substance use combined with rising temperatures.

4. Line 393-394: More sensitivity analyses on model parameters are needed to check the robustness of the results. For example, different degrees of freedom (df), lag days, and spline types.

We analysed the appropriateness of various model fit using the Akaike Information Criterion (AIC) values. We include a detailed analysis of the model fit in the Methods of the revised manuscript (PP. 8-9, Lines 235-242):

To select the optimal fit for the non-linear dlnm terms, models for alcohol-related disorders and substance-related disorders were fit separately using a variety of plausible degrees of freedom (dfs) to model the lag-specific exposure – response function (df_{var}), as well as the function of the association over the examined lags (df_{lag}). A range of 2 to 5 for df_{lag} were considered, along with between 3 and 4 for df_{var} . The optimal values were selected by choosing the combination of df_{lag} and df_{var} with the lowest Akaike Information Criteria (AIC) values.³⁷ The models with lowest AIC values for both causes were $df_{lag}=4$ and $df_{var}=3$.

5. Besides the studies that were already mentioned in the manuscript, is there any other evidence that supports a relationship between temperature and alcohol- and substance-related disorders? In addition to human studies, are there any related animal studies?

One of the primary reasons we were compelled to investigate the association between temperature and alcohol- and substance-related disorders is because there was previously very limited detailed evidence of this link, despite biological plausibility. We have added details of another study which used a selected subset of total United States hospital visits to identify a positive association with temperature and substance-related disorders in the revised manuscript (P. 4, Lines 99-101):

A study leveraging information on Optum hospital visits, a selected sub-set of total United States hospital visits based on insurance records, identified a positive association with temperature and substance-related disorders.¹⁷

We have also included in the Introduction of the revised manuscript evidence of the association between temperature and substance-related disorders in rats (P. 4, Lines 101-104):

Studies of rats have found that high ambient temperatures impact the acquisition of 3,4-methylenedioxymethamphetamine (MDMA) and methamphetamine, as well as dopamine self-administration.¹⁸⁻²¹

6. Supplementary Figures 3-9: The limits of the y-axis need to be reset so the central estimate and the error bars can be fully displayed (e.g., Figure S3, Sedatives – NYC – 1%).

We have adjusted these throughout the revised Supplementary Information. For reference, we have copied Supplementary Figure 3 below as an example:

Supplementary Figure 3. Percentage change in hospital visit rates by selected percentiles of temperature relative to minimal temperature (-30.1°C (-22.18°F)) for cannabis, cocaine, opioid, and sedative hospital visits by location in New York State, were each of the lag days (0 to 6 days before) at the quoted temperature percentile before hospital visit. Points show the point estimates and whiskers represent 95% confidence intervals.

Reviewer #3 (Remarks to the Author):

Thank you very much for letting me review this interesting work. The authors present a novel investigation on the association between ambient temperature and hospitalizations due to substance abuse and alcohol. I believe it is a timely contribution with relevant findings for public health. The topic of mental health in the context of climate change is considered today a hot topic, given the recent evidence on the direct and indirect connection between climate hazards and the consequences for mental health patients. The research presented is new, and the method used here is robust. I would only suggest the authors to clarify a few aspects on the interpretation of the findings and method.

We thank the Reviewer for the thoughtful and constructive suggestions. We have responded point-by-point to the Reviewer's questions and comments below.

Abstract

- "which controls for season and confounding bias" - this is not accurate, case-crossover designs inherently control for temporal trends (both seasonal and long term trends) and time-invariant confounders or those time-varying factors that change at a longer time scale than the association of interest.

We have updated this description in the Abstract of the revised manuscript (P. 2, Lines 22-23):

[...]a case-crossover design, which controls for ~~season and confounding bias~~ seasonal patterns, long-term trends, and non- or slowly varying confounders, [...]

- I would suggest the authors to avoid repeating the results in the conclusion and invest the space into providing a broader context - for example, the public health implications in relation to two relevant threats - climate change and poor mental health connected with the abuse of substances in the US.

We have removed the summary of the results in the Abstract and have added a sentence about the broader context in the revised manuscript (PP. 2-3, Lines 36-58):

Our work highlights how hospital visits from alcohol- and substance-related disorders up to a threshold, highlighting potential impacts of rising are currently impacted by elevated temperatures and could be further affected by rising temperatures resulting from a changing climate on mental change. Enhanced social infrastructure and health-related outcomes system interventions could mitigate these impacts.

Intro

- In the introduction, I would suggest the authors to highlight the synergistic effect between increasing variability due to climate change, and the rise of mental health problems due to accelerated socioeconomic trends and poor healthcare system that lead to the abuse of substances in (young) adults.

In response to the Reviewer's suggestion, we have highlighted both the direct and indirect pathways between rising temperature from climate change and substance use in the revised manuscript (PP. 3-4, Lines 77-88):

There are plausible direct behavioral and physiological pathways for a relationship between changes in temperature and alcohol- and substance-related disorders; increased consumption in warmer weather, more perspiration, and temperature-dependent efficacies of certain substances, such as opioids, may all contribute to changes in alcohol and substance use and how a human body reacts to their consumption.¹⁰ There are also indirect pathways between rising temperatures from climate change and worsening mental health outcomes that could lead to the increasing use of alcohol and other substances, such as deteriorating social fabric and widening inequality.^{11,12}

- In the aims, the authors state "how short term changes in exposure to daily temperature was associated with hospitalizations..." in this way, the readers would consider that the exposure variable is the "change in temperature" but it is actually not. The method here provides RRs that are measures of change in hospitalization risk based on different levels of temperature (18C vs. 10C).

We have updated this sentence in the revised manuscript (P. 5, Lines 123-125):

The aim of this study was to evaluate (a) how ~~short term changes in exposure to~~ daily temperature was associated with ~~hospitalizations~~ hospital visits due to alcohol- or substance-related disorders (including alcohol, cannabis, cocaine, opioids and sedatives), [...]

Results

- In the descriptive analysis, I would suggest the authors to provide (age-standardized) hospitalization rates by ZIP or across time (to complement the absolute number of hospitalizations). In the way how it is reported now, one cannot disentangle whether the spatial and temporal patterns are just driven by changes in the demographic structure of the population.

We agree with the Reviewer that understanding the context of our results is important. However, we did not use population values as inputs to any of the case crossover analyses. Since we unfortunately do not have information on patient IDs and cannot track individuals over time, estimating hospitalization rates with our data could lead to biased estimates. We are thus hesitant to include this information in our manuscript.

We have added further context in Introduction of the revised manuscript (P. 3, Lines 68-70):

In recent decades, there has been an increasing trend of heavy episodic drinking and alcohol-related morbidity and mortality in the United States, particularly in middle-aged to older adults.⁴

We have also added a section in the discussion of the revised manuscript which discusses the wider context of our results in the United States' recent history of alcohol and substance use (P. 15, Lines 592-604):

These results should be taken in the context of the past few decades that New York State and the United States have experienced. The opioid epidemic in the United States during the past few decades has resulted in a large increase in usage and dependency on prescription and illicitly-sourced opioids.⁴¹ Drug overdose deaths in the United States have increased more than five times since the end of the 20th century.⁴² Alcohol-attributable deaths during 2015-2019 have been estimated to represent 1 in 8 deaths among adults aged 20 to 64 years.³ The positive association of short-term temperature exposure and alcohol- and substance-related disorder hospital visits could further exacerbate negative alcohol- and substance-related outcomes in the United States with rising temperatures under climate change.

- Related to the comment above, I would avoid using "daily" increases of temperature - it would be prone to misunderstandings, since it seems that it refers to changes in temperature from one day to the other (e.g. 107 a daily increase from the daily average to the 90th percentile).

We have updated the language throughout to avoid referring to 'daily' increases of temperature in the revised manuscript (P. 11, Lines 310-314):

For alcohol-related disorder ~~hospitalizations, a daily~~ hospital visits, an increase in temperature from the period ~~daily average (10~~minimum (-30.1°C) (-22.2°F)) to the 75th percentile (18.48°C) (65.8°F)) across 0-6 lag days was associated with a cumulative 4.3%-24.6% (95%CI, 3.0%-5.5-14.6%-34.6%) increase in ~~daily hospitalization~~ hospital visit rates, and ~~a daily~~ an increase from the ~~daily~~ average to the 90th percentile (22.37°C) (72.9°F)) was associated with a cumulative 5.3-25.6% (95%CI, 3.5%-15.4%-35.7-0%) increase (Figure 3).

- Not sure whether the use of "increase in hospitalization rates" since in this context we are talking about increase in risk (e.g., line 107)

This is a case-crossover study design that uses incidence density sampling for the selection of control days. Therefore, the odds ratios from the conditional logistic models estimate rate ratios (Maclure 1991; Rothman et al. 2008).

- The authors mention in line 111 that " we did not find evidence of an association " with an estimate 1.2% (95%CI, -0.1%-2.4%) - it is true that the association estimate is smaller compared to the other estimate, and non-statistically significant. I would be more incline to interpret this finding as there seems to be an indication of an association although smaller and non-statistically significant.

We agree with the Reviewer that we were overly conservative in our interpretation. We have amended this, further noting that we have changed the reference point for the changes in hospital visit rates to the minimal temperature, in the revised manuscript (P. 11, Lines 316-319):

For substance-related disorders, we ~~did not find evidence~~ offound an association between ~~daily~~ increases in temperatures from ~~10~~the period minimum (-30.1°C (-22.2°F)) to the 75th percentile (18.48°C; (65.8°F)), with a ~~cumulative 1.2~~(38.8% (95%CI, -0.1% 2.4%) increase, nor from 10°C to 22.3°C, with a cumulative 1.4% (95%CI, -0.3% 3.1-28.7%-48.8%) increase.

- Related to the previous comment, I don't agree with the authors of using the mean temperature value as reference - why not using the 10th percentile or the minimum value? so most of the temperature range is above that value and it is easier to interpret. I would suggest the authors to first describe the shape in both cases - close to linear with a kind of plateau at very high temperatures. This shape has been found in other studies on mental health and temperature (suicides - Kin et al <https://doi.org/10.1289/EHP4898>. ; Barr et al 2022 doi:10.4414/SMW.2022.w30115).

We thank the Reviewer for this suggestion. We now quote the rate ratio for changes from the minimal temperature in the range (-30.1°C (-22.18°F); lowest risk temperature), e.g., (P. 11, Lines 310-314):

For alcohol-related disorder hospitalizations, a daily hospital visits, an increase in temperature from the period ~~daily average~~ ~~(10 minimum (-30.1°C) (-22.2°F))~~ to the 75th percentile (18.48°C) (65.8°F) across 0-6 lag days was associated with a cumulative ~~4.3%–24.6%~~ (95%CI, ~~3.0%–5.5~~14.6%–34.6%) increase in ~~daily hospitalization~~ hospital visit rates, and ~~a daily~~an increase from the ~~daily~~ average to the 90th percentile (22.37°C) (72.9°F) was associated with a cumulative ~~5.3~~25.6% (95%CI, ~~3.5%–15.4%–35.7~~7.0%) increase (Figure 3).

The revised Figure 3 can be seen below for reference:

Figure 3. Exposure-response ~~curve~~curves (top) of cumulative percentage change in ~~hospitalization~~hospital visit rates relative to ~~average~~minimal temperature ~~(10(-30.1°C) (-22.18°F))~~ for alcohol- and substance-related disorder ~~hospitalizations~~hospital visits, were each of the lag days (0 to 6 days before) at the quoted temperature before ~~hospitalization~~hospital visit. Black lines show the point estimates and orange ribbons represent 95% confidence intervals. ~~Histograms~~ (bottom) of temperature records for cases.

REVIEWERS' COMMENTS:

Reviewer #1 (Remarks to the Author):

The authors have done a nice job addressing reviewer comments.

Reviewer #2 (Remarks to the Author):

The manuscript has been greatly improved after the revisions. I have no further comments to provide. My appreciation goes to the authors for their intriguing and thought-provoking work.

Yiqun

Reviewer #3 (Remarks to the Author):

I would like to thank the authors for addressing my comments and providing appropriate responses to my suggestions.

I agree with the applied modifications. I would like to clarify my point regarding the use of standardised hospitalisation rates - it might not be appropriate to compare regions and time slices using the (crude) absolute incidence because the patterns might be driven by differences in demographic structure. In any case, I think it is a minor issue, and I believe with a clarification in the interpretation of the results (say that the observed patterns can be driven by differences in demographic structure of the population) would be enough.

We thank the Editors and Reviewers for their thoughtful and constructive suggestions. We have revised the manuscript in response to the Reviewers' comments, as detailed below.

All page/line/reference numbers refer to the tracked revised manuscript.

Reviewers' comments:

Reviewer #1 (Remarks to the Author):

The authors have done a nice job addressing reviewer comments.

We thank the Reviewer for the thoughtful and constructive comments.

Reviewer #2 (Remarks to the Author):

The manuscript has been greatly improved after the revisions. I have no further comments to provide. My appreciation goes to the authors for their intriguing and thought-provoking work.

Yiqun

We thank the Reviewer for the thoughtful and constructive comments.

Reviewer #3 (Remarks to the Author):

I would like to thank the authors for addressing my comments and providing appropriate responses to my suggestions.

We thank the Reviewer for the thoughtful and constructive suggestions. We have responded point-by-point to the Reviewer's questions and comments below.

I agree with the applied modifications. I would like to clarify my point regarding the use of standardised hospitalisation rates - it might not be appropriate to compare regions and time slices using the (crude) absolute incidence because the patterns might be driven by differences in demographic structure. In any case, I think it is a minor issue, and I believe with a clarification in the interpretation of the results (say that the observed patterns can be driven by differences in demographic structure of the population) would be enough.

We have updated this section in the Results section of the revised manuscript (PP. 11-12, Lines 235-241):

The maximal total number of hospital visits in a single ZIP Code was 6,479 for alcohol-related disorder hospital visit in Troy (12180) and 8,026 substance-related disorder hospital visits in East Harlem (10029-), though the observed patterns can be driven by differences in demographic structure of the population. Many cases were concentrated in urban environments. Overall, there was a high correlation ($R=0.98$) between total numbers of hospital visits for both alcohol- and substance-related disorders across all ZIP Codes.